# Clinical criteria to exclude acute vascular pathology on CT angiogram in patients with dizziness

Long H. Tu[1]*, Ajay Malhotra[1], Arjun K. Venkatesh[2], Richard A. Taylor[2], Kevin N. Sheth[3], Reza Yaesoubi[4], Howard P. Forman[1], Soundari Sureshanand[5], Dhasakumar Navaratnam[3]

1 Department of Radiology and Biomedical Imaging, Yale School of Medicine, New Haven, CT, United States of America, 2 Department of Emergency Medicine, Yale School of Medicine, New Haven, CT, United States of America, 3 Department of Neurology, Yale School of Medicine, New Haven, CT, United States of America, 4 Health Policy and Management, Yale School of Public Health, New Haven, CT, United States of America, 5 Yale Center for Clinical Investigation, Yale School of Medicine, New Haven, CT, United States of America

* long.tu@yale.edu

**Data Availability Statement:** Deposited files can be found at: DOI 10.17605/OSF.IO/HWUK2.

**Funding:** This work was supported by grant UL TR001863 from the National Center for Advancing

## Abstract

### Background

Patients presenting to the emergency department (ED) with dizziness may be imaged via CTA head and neck to detect acute vascular pathology including large vessel occlusion. We identify commonly documented clinical variables which could delineate dizzy patients with near zero risk of acute vascular abnormality on CTA.

### Methods

We performed a cross-sectional analysis of adult ED encounters with chief complaint of dizziness and CTA head and neck imaging at three EDs between 1/1/2014-12/31/2017. A decision rule was derived to exclude acute vascular pathology tested on a separate validation cohort; sensitivity analysis was performed using dizzy "stroke code" presentations.

### Results

Testing, validation, and sensitivity analysis cohorts were composed of 1072, 357, and 81 cases with 41, 6, and 12 instances of acute vascular pathology respectively. The decision rule had the following features: no past medical history of stroke, arterial dissection, or transient ischemic attack (including unexplained aphasia, incoordination, or ataxia); no history of coronary artery disease, diabetes, migraines, current/long-term smoker, and current/long-term anti-coagulation or anti-platelet medication use. In the derivation phase, the rule had a sensitivity of 100% (95% CI: 0.91–1.00), specificity of 59% (95% CI: 0.56–0.62), and negative predictive value of 100% (95% CI: 0.99–1.00). In the validation phase, the rule had a sensitivity of 100% (95% CI: 0.61–1.00), specificity of 53% (95% CI: 0.48–0.58), and negative predictive value of 100% (95% CI: 0.98–1.00). The rule performed similarly on dizzy

Translational Science. The funders had no role in study design, data collection and analysis, decision to publish, or preparation of the manuscript.

**Competing interests:** I have read the journal's policy and the authors of this manuscript have the following competing interests: LT reported receiving royalties for 2 textbooks, Search Pattern: A Systematic Approach to Diagnostic Imaging (2020) and A Brief Guide to the Neuroradiology Fellowship (2021), and was a student in the Yale University Investigative Medicine Program, which receives funding from the National Center for Advancing Translational Science, a component of the National Institutes of Health (NIH). KS reported receiving grants from Hyperfine during the conduct of the study and from the NIH, American Heart Association, and Biogen outside the submitted work. KS also reported receiving personal fees from Zoll (data and safety monitoring board chair), Alva Equity, and Cerovasc outside the submitted work. AV reported receiving grants from the US Centers for Medicare and Medicaid Services, Moore Foundation, American College of Radiology, and American College of Emergency Physicians outside the submitted work. RT reported receiving grants from the FDA, Moore Foundation, NIH, Society for Improving Diagnosis in Medicine (SIDM). No other disclosures were reported. The contents of this work are solely the responsibility of the authors and do not necessarily represent the official view of the NIH. This does not alter our adherence to PLOS ONE policies on sharing data and materials.

stroke codes and was more sensitive/predictive than all NIHSS cut-offs. CTAs for dizziness might be avoidable in 52% (95% CI: 0.47–0.57) of cases.

## Conclusions

A collection of clinical factors may be able to "exclude" acute vascular pathology in up to half of patients imaged by CTA for dizziness. These findings require further development and prospective validation, though could improve the evaluation of dizzy patients in the ED.

## Introduction

Posterior circulation stroke in patients presenting with dizziness is difficult to diagnose–delayed or missed diagnosis may occur in 37% of such cases on first medical contact [1]. An underlying ischemic stroke is present in 3–5% of patients presenting dizziness to the emergency department (ED) [2]. Large vessel occlusion (LVO) is the underlying etiology in approximately a third of posterior circulation ischemic strokes [3, 4]. LVO is therefore expected in ~1% of dizzy patients presenting to the ED. The National Institutes of Health Stroke Scale (NIHSS) is the most commonly used predictor in pre-hospital/pre-CTA risk assessment for LVO, however even a NIHSS of zero does not guarantee absence of LVO [5, 6].

Stroke or vessel occlusion with low NIHSS is more common in the posterior circulation and with non-focal presentations such as dizziness [5, 7]. Other acute vascular pathologies detectable on CTA include dissection and medium/small vessel occlusion; rarely, even ruptured aneurysm may initially present as dizziness [2, 8–10]. The early detection of these abnormalities may also impact patient management [11]. Therefore, predictive tools used to reassure against the need for emergent CTA or pre-hospital diversion to centers with endovascular capability would ideally consider these entities as well. In the absence of vessel abnormalities on CTA, underlying stroke is better detected by MRI or specialized bedside maneuvers [2, 12, 13]. Excluding the need for CTA may improve diagnosis by facilitating triage to these more sensitive modalities.

We hypothesize that there is a collection of clinical variables that can delineate a subpopulation of dizzy patients in whom there is a near zero probability of acute vascular pathology detectable on CTA. Such a collection of variables could form the basis of a decision rule guiding the selection of patients for CTA versus alternative testing. In this study, we perform a retrospective analysis to derive a decision rule excluding vascular pathology in dizzy ED patients. We then validate the rule on a temporally separate validation cohort and assess applicability to a similar group of "stroke code" presentations with dizziness. We compare performance to the NIHSS and estimate the proportion of CTA exams which are potentially avoidable.

## Methods

### Setting and design

We performed a cross-sectional analysis based on adult patient visits to one of three EDs in a healthcare system between 1/1/2014-12/31/2017. The first ED is a comprehensive stroke center, the second is a primary stroke center, the last is a smaller community ED. Adult (age ≥18 years) patient encounters with a chief complaint of dizziness who received CTA head and neck imaging were included for the analysis. Adult "stroke code" encounters during the same time period, with NIHSS ≤ 7 and a positive review of systems for dizziness were also obtained for a

sensitivity analysis in patients presenting within the treatment window with high suspicion of stroke. An NIHSS cut off 7 was chosen based on prior literature showing that patients with NIHSS > 7 have greater risk of LVO, independent of presenting with dizziness [6, 14]. Additional details of patient eligibility for CTA imaging are provided in Item 1 in S1 File.

The study was approved by the Yale University IRB with consent waived. The decision rule was developed with reference to the transparent reporting of a multivariable prediction model for individual prognosis or diagnosis (TRIPOD) guidelines [15].

## Data collection

Established vascular risk factors for stroke/LVO were identified based on a review of major recent literature and used to inform the first phase of feature selection [16–20] [Item 2 in S1 File]. Clinical data expected to be available at the time of an imaging order were extracted from the electronic medical record database (EPIC)–including but not limited to recognized risk factors. Demographic information, past medical history (PMH), review of systems (ROS), and physical exam (PE) findings were obtained. Past medical history data also captured categories of medication use, smoking, and other substance use. Physical exam findings included systems-based exam findings as well as NIHSS and Glasgow Coma Scale data. All associated CTA head and neck exam reports were obtained.

## Case categorization

CTA head and neck exam reports were categorized based on the presence of acute vascular abnormality. Acute vascular pathology was defined as large vessel occlusion, smaller arterial or venous occlusion, non-occlusive dissection, and aneurysm or other vascular lesion with hemorrhage. Operational definition of LVO is provided in Item 3 in S1 File. Various etiologies were included to account for the possibility that some patients with dizziness who do not have LVO may have other acute pathology whose acute management would be altered if detected on CTA. A decision rule excluding LVO but missing other entities requiring acute intervention would be less clinically useful. Age indeterminate though potentially acute findings were categorized as "positive" cases; this categorization was performed to maximize sensitivity even at the cost of potentially reduced specificity.

## Feature coding and missing data

One-hot encoding was used to covert PMH data into categorical variables. Instead of grouping past medical history elements into categories (e.g., history of cancer or history of diabetes), specific diagnosis codes were included, to assess differential risk that may arise from different severity or manifestations of medical conditions. ROS and PE findings were represented by two binary variables, one representing occurrence of a "pertinent positive" (e.g., positive review of systems for headache) the other when occurring as a "pertinent negative" (e.g., negative review of systems for nausea). Absent or missing documentation was therefore represented as a null value for both pertinent positive and pertinent negative variables. Numerical values (e.g., age, pain rating) were coded as continuous variables. Additional details of feature encoding are given in Item 4 in S1 File.

## Feature selection

The study population was split into training and validation cohorts consisting of the first 75% and last 25% of included patient encounters, respectively (corresponding to encounters before and after 14:27 on 5/12/2017). Features that were rarely documented in the medical record

and therefore impractical for use in a decision rule were excluded. We initially assessed the 200 most documented PMH features, 100 most commonly ROS features, all numerical vital signs, and the 100 most commonly PE features; these were drawn from an initial pool of 1500, 186, and 266 of PMH, ROS, and PE features and covered 74.5%, 96.9%, and 97.5% of instances of documentation in these categories respectively. Features were then filtered to remove those with low relationship to the target variable based on Chi-Squared test, leaving the 100 highest ranked features.

## Decision rule derivation via sequential covering

A decision tree was generated of sufficient depth to categorize all cases using the CART (Classification and Regression Trees) algorithm. Features were also ranked on random forest (Gini) importance to assess ability to separate positive and negative cases not just in the single tree, but across an ensemble of trees [21]; we used 100 trees, each with access to all top 100 Chi-Squared ranked features and allowed sufficient depth to categorize all cases. Features were then extracted to a decision rule in two phases. First, established vascular risk factors with high Gini importance (rank $\leq$ 5) were added to the list. Second, other (less-recognized) potential predictors were identified, based on highest Gini importance and location on the decision path predicting the largest leaf node without acute findings. Continuous variables were required to have a monotonic relationship with the target variable for inclusion, to account for the potential bias of Gini criteria toward high cardinality features [22]. Each time a feature was added to the list in either phase, the decision tree was re-created using cases not excluded by the evolving decision rule. Gini importance and decision tree structure were therefore used to obtain a list of features accounting for all positive instances of the target variable (i.e., via sequential covering [23]). Lastly, features on the list describing differing manifestations of a single condition were grouped into clinically coherent categories to arrive at the final decision rule. See Item 5 in S1 File for a schematic of this process.

## Decision rule validation and sensitivity analysis

The decision rule was applied to the temporally separated testing cohort, with performance compared to the NIHSS. A sensitivity analysis for hyperacute presentations was also performed, by applying the rule to "stroke code" presentations with dizziness (within the review of systems) and a NIHSS $\leq$ 7.

## Performance measures and statistical analysis

Sensitivity, specificity, positive predictive value (PPV), and negative predictive value (NPV) were calculated, with 95% confidence intervals (CI) using the Wilson score method [24]. The proportion of patients predicted to have a negative exam (potentially avoidable CTAs) was also reported with 95% CI using Wilson score [24].

## Software tools

Data management, coding, and analysis were done with Python (Version 3.7; Python Software Foundation; Delaware, US) in the PyCharm Integrated Development Environment (Version 2020.2.2; JetBrains; Prague, Czech Republic). Statistical and data analytic packages included pandas, sklearn, dtreeviz, and tableone [25]. Manual review and categorization of CTA findings was performed in Microsoft Excel (Version 2203; Microsoft; Redmond, Washington, US).

## Results

### Case characteristics (training and testing cohort)

During the study period, 15,483 adult patients presented to the ED with a chief complaint of dizziness. Of these, 1,429 (9.2%), received a CTA head and neck exam, and were included for analysis. Review of CTA head and neck results revealed 47 (3.3%) cases of acute vascular pathology (n = 31, 2.2% for LVO only) [Fig 1]. This set of cases was used to produce the training and validation cohorts. Demographic and clinical characteristics of patients with and without acute vascular pathology are provided in Table 1. Details of the positive cases are given in Item 6 in S1 File.

The 75/25 temporal split of the study population into training and validation cohorts generated with data sets with 1072 and 357 cases, respectively. There were 41 cases of acute vascular pathology in the training set and 6 in the validation set (28 and 3 cases of LVO respectively) [Fig 1].

### Decision rule list

Generation of a decision rule from the training cohort resulted in a collection of 23 features delineating a subset of encounters with zero cases of acute pathology. Summarizing these into clinical categories, we arrive at a decision rule predicting absence of acute vascular pathology in patients without a history of any of the following: stroke/transient ischemic attack (TIA),

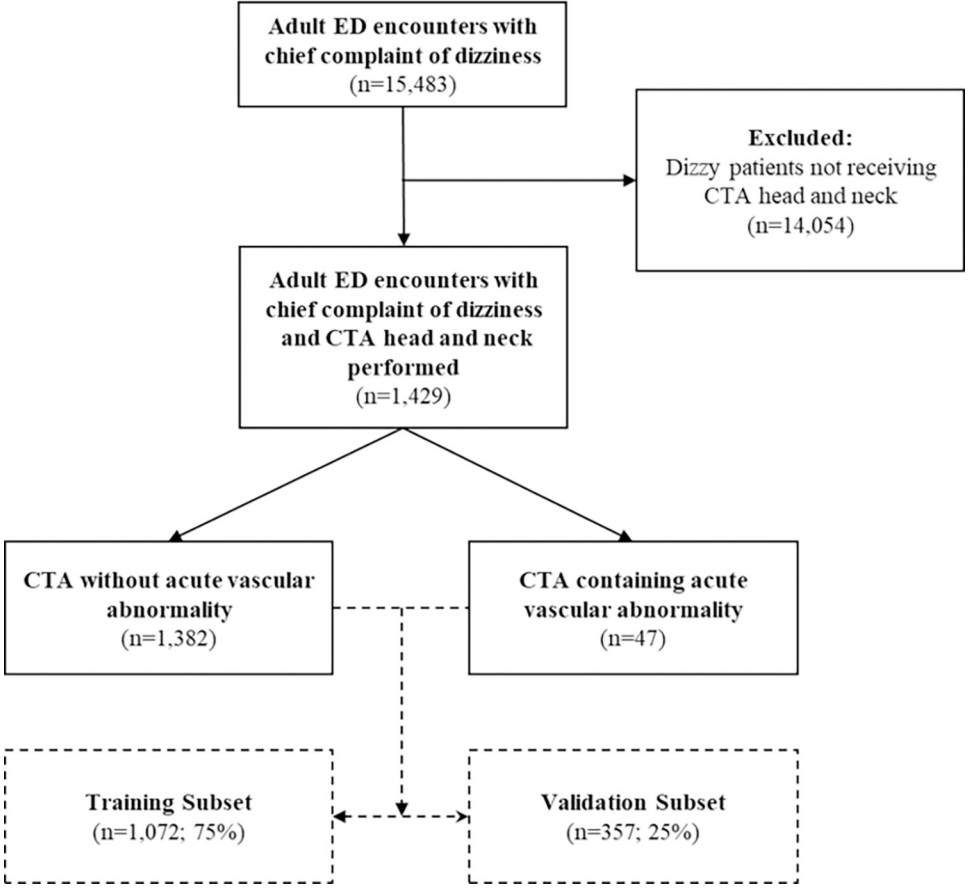

**Fig 1. Flow of patients to categorization of acute vascular abnormality on CTA and subsequent temporal separation of cases into a training and validation cohort.**

**Table 1. Demographic and characteristics of cases with and without acute vascular pathology.**

| Characteristic | Overall | No Acute Vascular Pathology | Acute Vascular Pathology | P-Value |
|---|---|---|---|---|
| Total Number of Cases | 1429 | 1382 | 47 | |
| Large Vessel Occlusion (LVO) | 31 (2.2) | | 31 (66.0) | |
| **Demographic Features** | | | | |
| Age, mean (SD) | 63.8 (15.6) | 63.6 (15.6) | 67.7 (15.1) | 0.079 |
| Male | 565 (39.5) | 542 (39.2) | 23 (48.9) | 0.235 |
| Ethnicity—Hispanic or Latino | 189 (13.2) | 185 (13.4) | 4 (8.5) | 0.452 |
| Race—American Indian or Alaska Native | 2 (0.1) | 2 (0.1) | 0 | 1 |
| Race—Asian | 31 (2.2) | 31 (2.2) | 0 | 0.622 |
| Race—Black or African American | 270 (18.9) | 259 (18.7) | 11 (23.4) | 0.539 |
| Race—Native Hawaiian or Other Pacific Islander | 6 (0.4) | 6 (0.4) | | 1 |
| Race—White or Caucasian | 975 (68.2) | 943 (68.2) | 32 (68.1) | 0.891 |
| Smoking—Active | 173 (12.1) | 167 (12.1) | 6 (12.8) | 0.931 |
| Smoking—Former | 560 (39.2) | 539 (39.0) | 21 (44.7) | 0.527 |
| **Past Medical History** | | | | |
| Coronary atherosclerosis of native coronary artery | 17 (1.2) | 14 (1.0) | 3 (6.4) | 0.016 |
| Presence of coronary angioplasty implant and graft | 35 (2.4) | 31 (2.2) | 4 (8.5) | |
| Atherosclerotic heart disease of native coronary artery without angina pectoris | 106 (7.4) | 98 (7.1) | 8 (17.0) | 0.019 |
| Atrial fibrillation | 29 (2.0) | 27 (2.0) | 2 (4.3) | 0.247 |
| Cerebral infarction, unspecified | 40 (2.8) | 35 (2.5) | 5 (10.6) | 0.009 |
| Dissection of vertebral artery | 9 (0.6) | 1 (0.1) | 8 (17.0) | <0.001 |
| Occlusion and stenosis of carotid artery without mention of cerebral infarction | 10 (0.7) | 7 (0.5) | 3 (6.4) | 0.003 |
| Transient ischemic attack (TIA), and cerebral infarction without residual deficits | 24 (1.7) | 20 (1.4) | 4 (8.5) | 0.007 |
| Personal history of transient ischemic attack (TIA), and cerebral infarction without residual deficits | 94 (6.6) | 89 (6.4) | 5 (10.6) | 0.231 |
| Essential (primary) hypertension | 599 (41.9) | 577 (41.8) | 22 (46.8) | 0.589 |
| Hyperlipidemia, unspecified | 276 (19.3) | 262 (19.0) | 14 (29.8) | 0.097 |
| Type 2 diabetes mellitus without complications | 178 (12.5) | 169 (12.2) | 9 (19.1) | 0.235 |
| Migraine, unspecified, not intractable, without status migrainosus | 22 (1.5) | 21 (1.5) | 1 (2.1) | 0.524 |
| Migraine, unspecified, without mention of intractable migraine without mention of status migrainosus | 12 (0.8) | 10 (0.7) | 2 (4.3) | 0.057 |
| Morbid obesity | 7 (0.5) | 6 (0.4) | 1 (2.1) | 0.209 |
| Long term (current) use of anticoagulants | 90 (6.3) | 81 (5.9) | 9 (19.1) | 0.002 |
| Long term (current) use of antithrombotics/antiplatelets | 27 (1.9) | 23 (1.7) | 4 (8.5) | 0.01 |
| Long term (current) use of aspirin | 173 (12.1) | 158 (11.4) | 15 (31.9) | <0.001 |

Unless otherwise specified, values are given as No. (%). Clinical characteristics that are known risk factors and that were the most commonly document features used for decision rule derivation are shown. Further details regarding the definition of current/former smoking and "long term" use of medications are given in Item 7 in S1 File.

unexplained speech difficulty/ataxia or visual disturbances (i.e., possible prior stroke/TIA), dissection, coronary artery disease, diabetes, migraines, active/long-term smoking, and current long-term anti-platelet or anti-coagulant use [Table 2].

## Decision rule performance

In the derivation phase, the rule applied to 603 of 1072 (56%) cases, excluding all 41 cases of pathology (28 cases of LVO). In the derivation set, the rule had a sensitivity of 100% (95% CI: 0.91–1.00), specificity of 59% (95% CI: 0.56–0.62), positive predictive value of 9% (95% CI:

**Table 2. List of features, summarized into clinical categories.**

| Broad Clinical Category | Features |
|---|---|
| Adult ED patient presenting with a chief complaint of dizziness<br>• Excluding those who may have an alternative indication for neurovascular imaging, aside from dizziness | ***Inclusion Criteria***: Adult ED patients with chief complaint of dizziness.<br>***Exclusion Criteria***: Other chief complaints, where dizziness is not the predominant concern, though may be an associated symptom (e.g., focal neurologic deficit, trauma, headache) |
| No PMH of cerebrovascular event, specifically:<br>• Dissection, stroke, or TIA (unexplained dysarthria/aphasia, ataxia, incoordination, and visual disturbances) | 1. PMH–Dissection of vertebral artery<br>2. PMH–Transient ischemic attack (TIA), and cerebral infarction without residual deficits<br>3. PMH–Other cerebral infarction<br>4. PMH–Aphasia<br>5. PMH–Dysarthria and anarthria<br>6. PMH–Ataxic gait<br>7. PMH–Ataxia, unspecified<br>8. PMH–Lack of coordination<br>9. PMH–Other lack of coordination<br>10. PMH–Visual Disturbances |
| No PMH of specific vascular risk factors:<br>• Coronary artery disease, diabetes, and current/long-term smoking, and migraines | 11. PMH—Coronary atherosclerosis of native coronary artery<br>12. PMH—Atherosclerotic heart disease of native coronary artery without angina pectoris<br>13. PMH—Presence of aortocoronary bypass graft<br>14. PMH—Type II or unspecified type diabetes mellitus without mention of complication, not stated as uncontrolled<br>15. PMH—Encounter for long-term (current) use of insulin<br>16. PMH—Long term (current) use of insulin<br>17. PMH—Nicotine dependence, cigarettes, uncomplicated<br>18. Smoking–active<br>19. PMH—Migraine, unspecified, without mention of intractable migraine without mention of status migrainosus<br>20. PMH—Migraine, unspecified, not intractable, without status migrainosus |
| No current specific long-term medication use:<br>• Anticoagulation or anti-platelet agents | 21. PMH—Encounter for long-term (current) use of aspirin<br>22. PMH—Long term (current) use of antithrombotics/antiplatelets<br>23. PMH—Long term (current) use of anticoagulants |

23 features were iteratively extracted from the training dataset, which together predict absence of acute vascular pathology in patients presenting to the ED with dizziness. Along with study inclusion/exclusion criteria, these are grouped in four broad groups. Further details regarding the definition of current/former smoking and "long term" use of medications are given in Item 7 in S1 File.

0.07–0.12), and negative predictive value of 100% (95% CI: 0.99–1.00). A two-by-two table of patients excluded (vs. not excluded) by the decision rule and those with (vs. without) acute vascular pathology in the derivation phase is given in Item 8 in S1 File.

Application of the decision rule to the validation cohort predicted absence of acute vascular pathology in 185 of 357 (52%) cases, excluding all 6 cases of pathology (3 cases of LVO). This corresponds to a sensitivity of 100.0% (95% CI: 0.61–1.00), specificity of 53% (95% CI: 0.48–0.58), positive predictive value of 4% (95% CI: 0.02–0.07), and negative predictive value of 100% (95% CI: 0.98–1.00). The proportion of patients with near-zero risk and potentially avoidable CTAs was 52% (CI: 0.47–0.57).

**Table 3. NIHSS and decision rule performance.**

| Test | Sensitivity | Specificity | Positive Predictive Value (PPV) | Negative Predictive Value (NPV) | Predicted Negative (PN) |
|---|---|---|---|---|---|
| NIHSS $\leq 8$ | 0.00 (0.00–0.43) | 0.03 (0.01–0.11) | 0.00 (0.00–0.06) | 0.29 (0.08–0.64) | 0.10 (0.05–0.20) |
| NIHSS $\leq 7$ | 0.00 (0.00–0.43) | 0.03 (0.01–0.11) | 0.00 (0.00–0.06) | 0.29 (0.08–0.64) | 0.10 (0.05–0.20) |
| NIHSS $\leq 6$ | 0.00 (0.00–0.43) | 0.05 (0.02–0.13) | 0.00 (0.00–0.06) | 0.38 (0.14–0.69) | 0.12 (0.06–0.22) |
| NIHSS $\leq 5$ | 0.20 (0.04–0.62) | 0.08 (0.04–0.18) | 0.02 (0.00–0.09) | 0.56 (0.27–0.81) | 0.13 (0.07–0.23) |
| NIHSS $\leq 4$ | 0.20 (0.04–0.62) | 0.10 (0.05–0.18) | 0.02 (0.00–0.09) | 0.60 (0.31–0.83) | 0.15 (0.08–0.25) |
| NIHSS $\leq 3$ | 0.20 (0.04–0.62) | 0.13 (0.07–0.23) | 0.02 (0.00–0.10) | 0.67 (0.39–0.86) | 0.18 (0.11–0.29) |
| NIHSS $\leq 2$ | 0.40 (0.12–0.77) | 0.18 (0.10–0.29) | 0.04 (0.01–0.13) | 0.79 (0.52–0.92) | 0.21 (0.13–0.32) |
| NIHSS $\leq 1$ | 0.40 (0.12–0.77) | 0.29 (0.19–0.41) | 0.04 (0.01–0.15) | 0.86 (0.65–0.95) | 0.31 (0.22–0.43) |
| NIHSS = 0 | 0.60 (0.23–0.88) | 0.53 (0.41–0.65) | 0.09 (0.03–0.24) | 0.94 (0.81–0.98) | 0.52 (0.40–0.64) |
| Decision Rule (validation) | 1.00 (0.61–1.00) | 0.53 (0.48–0.58) | 0.04 (0.02–0.07) | 1.00 (0.98–1.00) | 0.52 (0.47–0.57) |

Sensitivity and specificity of NIHSS cut-offs (0–8) and the clinical decision rule, when applied to the validation cohort. 95% confidence intervals are listed in parentheses. The clinical decision rule had higher sensitivity, specificity, and NPV than all cut-off values of the NIHSS; it also identified a larger proportion of patients (predicted negative) in whom CTA might be avoidable.

The separate cohort used for sensitivity analysis ("stroke codes," dizziness, NIHSS $\leq 7$) consisted of 81 cases with 12 acute vascular findings (10 LVO). Application of the decision rule predicted absence of acute pathology in 35 patients (43%), excluding all cases of acute pathology. Sensitivity was 1.00 (95% CI: 0.76–1.00), specificity 0.51 (95% CI: 0.39–0.62), positive predictive value 26% (95% CI: 0.16–0.40), and negative predictive value was 100% (95% CI: 0.90–1.00). The proportion of dizzy "stroke code" presents with near-zero risk and potentially avoidable CTAs was 43% (95% CI: 0.33–0.54).

Performance of the decision rule in the validation phase compared to NIHSS cut-offs is given in Table 3. Results of the derivation phase and sensitivity analysis are provided separately [Item 9 in S1 File]. Details of NIHSS documentation and performance are given in the Item 10 in S1 File.

## Discussion

In this study, we used retrospective data and a sequential covering approach to derive a decision rule excluding large vessel occlusion and other vascular pathology in patients presenting with dizziness. The resultant rule consists of well-recognized vascular risk factors and as well as other features identified based on random forest importance. The decision rule predicts absence of acute vascular pathology in patients who do not have any of the following past medical history features:

*Stroke/transient ischemic attack (TIA), unexplained speech difficulty/ataxia or visual disturbances (i.e., possible prior stroke/TIA), dissection, coronary artery disease, diabetes, migraines, active/long-term smoking, and current long-term anti-platelet or anti-coagulant use.* See Fig 2 for a visual representation of the rule.

In the testing cohort, the decision rule excluded acute vascular pathology with 100% sensitivity and 53% specificity, The rule was more sensitive than conventionally used NIHSS thresholds (NIHSS $\leq 7$), as well as NIHSS = 0. Similar performance of the rule, 100% sensitivity and 51% specificity, was seen when applied to "stroke code" presentations with dizziness and with relatively lower risk of anterior circulation LVO (NIHSS $\leq 7$). High sensitivity suggests utility as a "rule out" tool; non-applicability of the rule does necessarily indicate need for CTA or

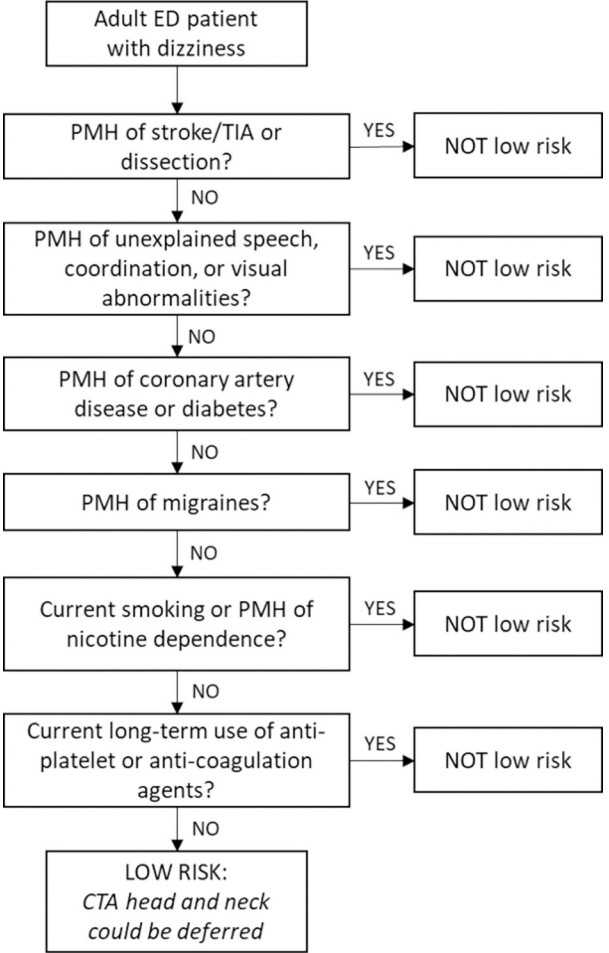

**Fig 2. Visualization of a potential decision rule to exclude acute vascular pathology in adult ED patients with chief complaint of dizziness.** Clinical criteria are grouped into closely related categories. Further development and validation are needed prior to application in clinical contexts.

presence of underlying pathology. The decision rule applied to 52% of cases in the validation cohort and 43% of cases in the sensitivity analysis, suggesting that nearly half of CTA head and neck examinations might be avoidable, if such a tool were sufficiently reassuring to defer neurovascular imaging.

Dizziness is a non-specific symptom that accompanies a wide range of neurologic, cardiovascular, psychiatric, and other disease processes [2]. In this study, we focused on those patients in whom dizziness is the presenting complaint, and in whom there may be no alternative presentations warranting neurovascular evaluation. For example, patients who are dizzy in the setting of trauma or thunderclap headache would receive neurovascular evaluation for other reasons.

Our decision rule suggests that documented (otherwise unexplained) speech, coordination, and vision abnormalities be considered potential prior stroke or TIA for the purposes of risk stratification. This would be consistent with prior research showing that as many as 90% of posterior circulation TIAs may be misdiagnosed on first medical contact [26]. Among established vascular risk factors, smoking status and diabetes were included in the decision rule; hypertension and hyperlipidemia were conspicuously absent. It is conceivable that these or

other risk factors for stroke would provide predictive value if the rule were updated or validated with additional data sets. Medical history elements seemed to be much more strongly predictive than review of system or physical exam findings, which may reflect the pre-selection of ED visits with dizziness as a primary concern. A notable difference between the risk factors highlighted by our study and existing predictive tools is the inclusion of migraine history as a predictive factor. Migraine with aura is a well-described, though perhaps underappreciated, risk factor for both dissection and stroke [27–29]. Our study suggests potential value within prediction tools alongside more established vascular disease processes.

Prior research suggests that vascular imaging to evaluate for large vessel occlusion may be cost-effective in acute minor stroke patients even when the risk of LVO is low (0.2%) [30]. Predictive tools for vascular pathology in stroke need to have near-perfect sensitivity to appropriately defer neurovascular evaluation. The collection of risk factors highlighted by our study suggests that subpopulations of patients with dizziness may exist with near-zero risk for pathology detectable on CTA. None of the patients excluded by the decision rule (in the training, validation, or sensitivity analysis cohorts) had an acute abnormality on CTA. Future research could assess the potential role of such a decision rule in conjunction with existing bedside exams and risk stratification tools.

Strokes in the posterior circulation are less often caused by LVO than by cardiac embolism or lacunar mechanism [3]. The decision rule developed in this work may assist in distinguishing underlying etiologies by excluding patients with stenotic atherosclerosis/vasculopathy who subsequently develop vessel occlusion. CTA is less sensitive for subtle ischemic events that arise from distal embolic and other etiologies [2]. Patients with minimal risk for LVO, but still suspected to have stroke might could be prioritized for more sensitive MRI, HINTS (head impulse, nystagmus, and test of skew), or subspecialist consultation. Vascular imaging could be deferred in this cohort and performed only if alternative indications arise.

## Limitations

Our study has some limitations. The first of these is the retrospective nature of data collection. Documentation in ED medical records is known to be inconsistent [31], which limits incorporation of predictive factors not appearing in the medical chart. Decision rule performance would also be impacted by medical history elements which may be present but not documented. Prospective and external validation would help address these limitations. Our data precede the COVID-19 pandemic; therefore, any potential thromboembolic risk from COVID-19 infection or related to rare vaccination side effects [32] are not captured. This possibility could be incorporated in future studies. Considering age-indeterminate findings on CTA to be potentially acute allowed us to err on the side of maximizing sensitivity, though could have reduced specificity for detection of acute abnormality. The consideration of a large and varied set of features allowed us to identify risk factors which may have been underappreciated, though could also reduce generalizability. The derivation of predictive tools for dizzy patients is complicated by the low prevalence of acute abnormality on imaging and therefore fewer positive events. We expect the need to refine the decision rule and estimates of performance with subsequent work using additional datasets. Most of the study limitations would have reduced the ability to identify patients at low risk; yet, we still found that half of patients may not have required CTA.

## Conclusion

Our study suggests that a collection of clinical variables may delineate a sizeable subpopulation of dizzy patients receiving CTA in whom there is a near zero risk of acute vascular pathology

including LVO. A decision rule composed of these predominantly well-recognized risk factors could apply to as many as half of patients currently evaluated by head and neck CTA. Further development and validation of this predictive tool would help guide efficient diagnosis and management of dizzy patients in the emergency department.

## Supporting information

**S1 File.**
(DOCX)

## Author Contributions

**Conceptualization:** Long H. Tu, Ajay Malhotra, Arjun K. Venkatesh, Richard A. Taylor, Kevin N. Sheth, Reza Yaesoubi, Howard P. Forman, Dhasakumar Navaratnam.

**Data curation:** Long H. Tu, Richard A. Taylor, Soundari Sureshanand.

**Formal analysis:** Long H. Tu, Arjun K. Venkatesh, Reza Yaesoubi.

**Funding acquisition:** Long H. Tu.

**Investigation:** Long H. Tu, Ajay Malhotra, Arjun K. Venkatesh, Richard A. Taylor.

**Methodology:** Long H. Tu, Arjun K. Venkatesh, Richard A. Taylor, Soundari Sureshanand.

**Project administration:** Ajay Malhotra, Howard P. Forman.

**Resources:** Richard A. Taylor, Soundari Sureshanand.

**Software:** Reza Yaesoubi.

**Supervision:** Ajay Malhotra, Arjun K. Venkatesh, Richard A. Taylor, Kevin N. Sheth, Reza Yaesoubi, Howard P. Forman, Dhasakumar Navaratnam.

**Validation:** Long H. Tu, Richard A. Taylor, Reza Yaesoubi, Dhasakumar Navaratnam.

**Visualization:** Long H. Tu, Reza Yaesoubi.

**Writing – original draft:** Long H. Tu.

**Writing – review & editing:** Long H. Tu, Ajay Malhotra, Arjun K. Venkatesh, Richard A. Taylor, Kevin N. Sheth, Reza Yaesoubi, Howard P. Forman, Soundari Sureshanand, Dhasakumar Navaratnam.

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
