## [Decision Letter · Decision Letter 0]

8 Nov 2022

PONE-D-22-23945Clinical Criteria to Exclude Acute Vascular Pathology on CT Angiogram in Patients with DizzinessPLOS ONE

Dear Dr. Tu,

Thank you for submitting your manuscript to PLOS ONE. After careful consideration, we feel that it has merit but does not fully meet PLOS ONE’s publication criteria as it currently stands. Therefore, we invite you to submit a revised version of the manuscript that addresses the points raised during the review process.

ACADEMIC EDITOR: All issues raised by expert reviewers are required. I recommend inclusion of criteria on how to judje application of *contrast* CTA head and neck exam (clinical values of biochemical data on renal function, etc.) to 3 institutes studied in supplementary data. I understand CTA is easy to test in ED in the US; however in some other countries (e.g., Germany, Japan), non-contrast MR diffusion images (<30sec) and MRA (~30 min) could be proceeded to detect LVO in the posterior circulation for EVT. 

We look forward to receiving your revised manuscript.

Kind regards,

Tatsushi Mutoh

Academic Editor

PLOS ONE

Journal Requirements:

“I have read the journal's policy and the authors of this manuscript have the following competing interests: LT reported receiving royalties for 2 textbooks, Search Pattern: A Systematic Approach to Diagnostic Imaging (2020) and A Brief Guide to the Neuroradiology Fellowship (2021), and was a student in the Yale University Investigative Medicine Program, which receives funding from the National Center for Advancing Translational Science, a component of the National Institutes of Health (NIH). KS reported receiving grants from Hyperfine during the conduct of the study and from the NIH, American Heart Association, and Biogen outside the submitted work. KS also reported receiving personal fees from Zoll (data and safety monitoring board chair), Alva Equity, and Cerovasc outside the submitted work. AV reported receiving grants from the US Centers for Medicare and Medicaid Services, Moore Foundation, American College of Radiology, and American College of Emergency Physicians outside the submitted work. No other disclosures were reported. The contents of this work are solely the responsibility of the authors and do not necessarily represent the official view of the NIH.”

Reviewers' comments:

Reviewer's Responses to Questions

**Comments to the Author**

1. Is the manuscript technically sound, and do the data support the conclusions?

Reviewer #1: Yes

Reviewer #2: Yes

2. Has the statistical analysis been performed appropriately and rigorously? 

Reviewer #1: Yes

Reviewer #2: I Don't Know

3. Have the authors made all data underlying the findings in their manuscript fully available?

Reviewer #1: No

Reviewer #2: Yes

4. Is the manuscript presented in an intelligible fashion and written in standard English?

Reviewer #1: Yes

Reviewer #2: Yes

5. Review Comments to the Author

Reviewer #1: The paper entilted "Clinical Criteria to Exclude Acute Vascular Pathology on CT Angiogram in Patients with

Dizziness" is well written. I have no concern about the methods, results, and data interpretations. There is only one minor issue need the authors to revise:

“Review of system” was incorrectly abbreviated as “PE” in the paragraph of data collection.

Reviewer #2: The purpose of this retrospective cross-sectional study was to identify a subpopulation of dizzy patients with near zero probability of acute vascular pathology detectable on CTA. The authors created a decision rule consisting of clinical factors that could exclude all cases of acute vascular pathology on CTA. The proposed sensitivity of the decision rule (100%) appears to be better than that of any thresholds of NIHSS scores up to 8 points.

This study documents interesting features concerning that as many as half of currently conducted head and neck CTA was avoidable.

I agree with the authors that this decision rule which was made to maximize sensitivity can potentially be a useful rule-out tool to detect dizzy patients who don’t need to undergo CTA, with future research assessment.

However, I have some suggestions for clarifying some aspects of the manuscript.

Specific points to be addressed are as follows.

1) Abstract (conclusion):

It is questionable to conclude that, "A collection of clinical factors may be able to “exclude” acute vascular pathology in up to half of the patients presenting to the ED with dizziness" since this study has excluded dizzy patients not receiving CTA head and neck, and targets acute vascular pathology detected on CTA. I recommend that the authors should tone down the statement as written in the discussion/conclusion of the manuscript and clarify that "acute vascular pathology” means that detectable on CTA.

2) Methods:

(3rd Para) Data Collection

(2nd page of the methods, 1st line) The authors wrote that “review of systems (PE)”, but isn’t it ROS?

3) Results:

(3rd Para) Decision Rule Performance

(4th page of the results, 5th line) The authors described the sensitivity and specificity of the decision rules applied in the derivation phase cohort. Since the rule was composed of negative forms and made as exclusion criteria, sensitivity and specificity seem complicated at first sight. Adding the conventional two-by-two table of patients with/without acute vascular pathology on CTA and patients included in the rules (low risk)/NOT included in the rules (NOT low risk), as Supplemental data would be helpful for readers.

4)Table1, Table2:

There are some criteria with “long-term”, such as long-term smoking, and long-term medication use. How long do you define the length of “long-term”? Please describe.

5)Table 3:

The table compares the sensitivity and specificity of NIHSS and the clinical decision rule when applied to the validation cohort. However, the performance of the decision rule during the derivation phase and sensitivity analysis are also added. I think that those are not appropriate to add in parallel in this table since they are applied in the different cohorts. I would recommend the values of NIHSS and validation cohort and those of derived phase and sensitivity analysis are presented separately.

6. PLOS authors have the option to publish the peer review history of their article (what does this mean?). If published, this will include your full peer review and any attached files.

Reviewer #1: No

Reviewer #2: No

---

## [Author Response · Author response to Decision Letter 0]

25 Nov 2022

*The following has been uploaded as a file with the revision, pasted below from Word file.*

We would first like to thank the academic editor and reviewers for their thorough and insightful comments. We have addressed all suggestions, and are amenable to further revisions should they help clarify this work.

ONE-D-22-23945

Clinical Criteria to Exclude Acute Vascular Pathology on CT Angiogram in Patients with Dizziness

PLOS ONE

Dear Dr. Tu,

Thank you for submitting your manuscript to PLOS ONE. After careful consideration, we feel that it has merit but does not fully meet PLOS ONE’s publication criteria as it currently stands. Therefore, we invite you to submit a revised version of the manuscript that addresses the points raised during the review process.

ACADEMIC EDITOR: All issues raised by expert reviewers are required. I recommend inclusion of criteria on how to judje application of contrast CTA head and neck exam (clinical values of biochemical data on renal function, etc.) to 3 institutes studied in supplementary data. I understand CTA is easy to test in ED in the US; however in some other countries (e.g., Germany, Japan), non-contrast MR diffusion images (<30sec) and MRA (~30 min) could be proceeded to detect LVO in the posterior circulation for EVT.

Thank you so much for this thoughtful perspective. We have elaborated on the selection of patients for contrast CTA head and neck, including with regard to renal function. We very much agree that where available, non-contrast MR diffusion and/or MRA offers the potential for alternative workflows – we have also provided additional discussion on these scenarios. The additions are within S1 Supporting Information and referenced in Methods (Setting and Design). The Supporting Information is re-numbered/reformatted throughout.

We look forward to receiving your revised manuscript.

Kind regards,

Tatsushi Mutoh

Academic Editor

PLOS ONE

Journal Requirements:

Done. Thank you.

“I have read the journal's policy and the authors of this manuscript have the following competing interests: LT reported receiving royalties for 2 textbooks, Search Pattern: A Systematic Approach to Diagnostic Imaging (2020) and A Brief Guide to the Neuroradiology Fellowship (2021), and was a student in the Yale University Investigative Medicine Program, which receives funding from the National Center for Advancing Translational Science, a component of the National Institutes of Health (NIH). KS reported receiving grants from Hyperfine during the conduct of the study and from the NIH, American Heart Association, and Biogen outside the submitted work. KS also reported receiving personal fees from Zoll (data and safety monitoring board chair), Alva Equity, and Cerovasc outside the submitted work. AV reported receiving grants from the US Centers for Medicare and Medicaid Services, Moore Foundation, American College of Radiology, and American College of Emergency Physicians outside the submitted work. No other disclosures were reported. The contents of this work are solely the responsibility of the authors and do not necessarily represent the official view of the NIH.”

We have added the relevant statement. Updated Competing Interests are included in the cover letter. 

Done. Thank you.

Thank you, yes. We will be providing repository information with acceptance.

We have done this. There is only one supporting information file, containing 10 items. Please feel free to edit/remove the details provided.

Done. Thank you.

Reviewers' comments:

Reviewer's Responses to Questions

Comments to the Author

1. Is the manuscript technically sound, and do the data support the conclusions?

Reviewer #1: Yes

Reviewer #2: Yes

2. Has the statistical analysis been performed appropriately and rigorously?

Reviewer #1: Yes

Reviewer #2: I Don't Know

3. Have the authors made all data underlying the findings in their manuscript fully available?

Reviewer #1: No

Reviewer #2: Yes

4. Is the manuscript presented in an intelligible fashion and written in standard English?

Reviewer #1: Yes

Reviewer #2: Yes

5. Review Comments to the Author

Reviewer #1: The paper entilted "Clinical Criteria to Exclude Acute Vascular Pathology on CT Angiogram in Patients with

Dizziness" is well written. I have no concern about the methods, results, and data interpretations. There is only one minor issue need the authors to revise:

“Review of system” was incorrectly abbreviated as “PE” in the paragraph of data collection.

Thank you so much for the consideration. Yes, this was an oversight. We have made the correction.

Reviewer #2: The purpose of this retrospective cross-sectional study was to identify a subpopulation of dizzy patients with near zero probability of acute vascular pathology detectable on CTA. The authors created a decision rule consisting of clinical factors that could exclude all cases of acute vascular pathology on CTA. The proposed sensitivity of the decision rule (100%) appears to be better than that of any thresholds of NIHSS scores up to 8 points.

This study documents interesting features concerning that as many as half of currently conducted head and neck CTA was avoidable.

I agree with the authors that this decision rule which was made to maximize sensitivity can potentially be a useful rule-out tool to detect dizzy patients who don’t need to undergo CTA, with future research assessment.

However, I have some suggestions for clarifying some aspects of the manuscript.

Specific points to be addressed are as follows.

1) Abstract (conclusion):

It is questionable to conclude that, "A collection of clinical factors may be able to “exclude” acute vascular pathology in up to half of the patients presenting to the ED with dizziness" since this study has excluded dizzy patients not receiving CTA head and neck, and targets acute vascular pathology detected on CTA. I recommend that the authors should tone down the statement as written in the discussion/conclusion of the manuscript and clarify that "acute vascular pathology” means that detectable on CTA.

Yes. We agree completely. The original statement was inexact. We have updated the relevant line in the abstract and clarified a similar line in the discussion/conclusion.

2) Methods:

(3rd Para) Data Collection

(2nd page of the methods, 1st line) The authors wrote that “review of systems (PE)”, but isn’t it ROS?

Yes. This was an error. Thank you. We have made the correction.

3) Results:

(3rd Para) Decision Rule Performance

(4th page of the results, 5th line) The authors described the sensitivity and specificity of the decision rules applied in the derivation phase cohort. Since the rule was composed of negative forms and made as exclusion criteria, sensitivity and specificity seem complicated at first sight. Adding the conventional two-by-two table of patients with/without acute vascular pathology on CTA and patients included in the rules (low risk)/NOT included in the rules (NOT low risk), as Supplemental data would be helpful for readers.

This is a great point. We have added such a table to S1 Supporting Information and referenced it in-text in the results.

4)Table1, Table2:

There are some criteria with “long-term”, such as long-term smoking, and long-term medication use. How long do you define the length of “long-term”? Please describe.

Thank you. We have described these designations further in S1 Supporting Information, with reference within the legends for Table 1 and Table 2. We would also like to note here that we noticed a typo in table 2 in enumerating 22 features summarized into 4 broad clinical categories. There are 23 features (two related to insulin use were previously listed on the same line) – we have made appropriate corrections in the legend/text. Please let us know if this is alright.

5)Table 3:

The table compares the sensitivity and specificity of NIHSS and the clinical decision rule when applied to the validation cohort. However, the performance of the decision rule during the derivation phase and sensitivity analysis are also added. I think that those are not appropriate to add in parallel in this table since they are applied in the different cohorts. I would recommend the values of NIHSS and validation cohort and those of derived phase and sensitivity analysis are presented separately.

Yes. We agree completely. We have removed the derivation and sensitivity analysis from this table. We have instead presented decision rule performance in the differing phases of the study in a table within the S1 Supporting Information, indicating that the measures are produced using differing patient cohorts. Thank you!

6. PLOS authors have the option to publish the peer review history of their article (what does this mean?). If published, this will include your full peer review and any attached files.

Do you want your identity to be public for this peer review? For information about this choice, including consent withdrawal, please see our Privacy Policy.

Reviewer #1: No

Reviewer #2: No

 Done.

---

## [Decision Letter · Decision Letter 1]

8 Jan 2023

Clinical Criteria to Exclude Acute Vascular Pathology on CT Angiogram in Patients with Dizziness

PONE-D-22-23945R1

Dear Dr. Tu,

We’re pleased to inform you that your manuscript has been judged scientifically suitable for publication and will be formally accepted for publication once it meets all outstanding technical requirements.

Kind regards,

Tatsushi Mutoh

Academic Editor

PLOS ONE

Additional Editor Comments (optional):

The authors have done a good job of addressing most of the comments from the reviewers.

I could only find some errors and/or corrections--please see two reviewers' comments.

Reviewers' comments:

Reviewer's Responses to Questions

**Comments to the Author**

1. If the authors have adequately addressed your comments raised in a previous round of review and you feel that this manuscript is now acceptable for publication, you may indicate that here to bypass the “Comments to the Author” section, enter your conflict of interest statement in the “Confidential to Editor” section, and submit your "Accept" recommendation.

Reviewer #1: All comments have been addressed

Reviewer #2: All comments have been addressed

2. Is the manuscript technically sound, and do the data support the conclusions?

Reviewer #1: Yes

Reviewer #2: Yes

3. Has the statistical analysis been performed appropriately and rigorously? 

Reviewer #1: Yes

Reviewer #2: I Don't Know

4. Have the authors made all data underlying the findings in their manuscript fully available?

Reviewer #1: Yes

Reviewer #2: Yes

5. Is the manuscript presented in an intelligible fashion and written in standard English?

Reviewer #1: Yes

Reviewer #2: Yes

6. Review Comments to the Author

Reviewer #1: The authors addressed all the concerns raised in the last round of review.

However, it seems that Table 3 was not revised correctly according to the suggestion from reviewer 2 (the derivation and sensitivity analysis should be removed).

Reviewer #2: Thank you for the revised manuscript. The authors answered the points raised by the reviewer.

Although I am mostly satisfied with this revision, I still find several minor corrections as follows.

1, Supporting information, item 7, line 4

There remain the words `Our stu`. Please delete them.

2, Table 3

In the rebuttal letter, the authors wrote that they had removed the derivation and sensitivity analysis from the table, but they are still in the table. Please check and correct them.

7. PLOS authors have the option to publish the peer review history of their article (what does this mean?). If published, this will include your full peer review and any attached files.

Reviewer #1: No

Reviewer #2: No

---

## [Editor Report · Acceptance letter]

27 Feb 2023

PONE-D-22-23945R1 

Clinical Criteria to Exclude Acute Vascular Pathology on CT Angiogram in Patients with Dizziness 

Dear Dr. Tu:

I'm pleased to inform you that your manuscript has been deemed suitable for publication in PLOS ONE. Congratulations! Your manuscript is now with our production department. 

Kind regards, 

on behalf of

Dr. Tatsushi Mutoh 

Academic Editor

PLOS ONE